# Effect of sulfur on sound velocity of liquid iron under Martian core conditions

Keisuke Nishida [1,9 ✉], Yuki Shibazaki [2,3], Hidenori Terasaki[4], Yuji Higo[5], Akio Suzuki[6], Nobumasa Funamori[7] & Kei Hirose [1,8]

Sulfur has been considered to be a predominant light element in the Martian core, and thus the sound velocity of Fe-S alloys at relevant high pressure and temperature is of great importance to interpret its seismological data. Here we measured the compressional sound velocity ($V_P$) of liquid Fe, $Fe_{80}S_{20}$ and $Fe_{57}S_{43}$ using ultrasonic pulse-echo overlap method combined with a Kawai-type multi-anvil apparatus up to 20 GPa, likely corresponding to the condition at the uppermost core of Mars. The results demonstrate that the $V_P$ of liquid iron is least sensitive to its sulfur concentration in the Mars' whole core pressure range. The comparison of seismic wave speeds of Fe-S liquids with future observations will therefore tell whether the Martian core is molten and contains impurity elements other than sulfur.

[1] Department of Earth and Planetary Science, The University of Tokyo, 7-3-1 Hongo, Bunkyo, Tokyo 113-0033, Japan. [2] Frontier Research Institute for Interdisciplinary Sciences, Tohoku University, 6-3 Aoba, Aramaki, Aoba, Sendai 980-8578, Japan. [3] International Center for Young Scientists, National Institute for Materials Science, 1-1 Namiki, Tsukuba, Ibaraki 305-0044, Japan. [4] Department of Earth and Space Science, Osaka University, 1-1 Machikaneyama-cho, Toyonaka, Osaka 560-0043, Japan. [5] Japan Synchrotron Radiation Research Institute, 1-1-1 Kouto, Sayo-cho, Sayo, Hyogo 679-5198, Japan. [6] Department of Earth Science, Tohoku University, 6-3 Aoba, Aramaki, Aoba, Sendai 980-8578, Japan. [7] Institute of Materials Structure Science, High Energy Accelerator Research Organization (KEK), 1-1 Oho, Tsukuba, Ibaraki 305-0801, Japan. [8] Earth-Life Science Institute, Tokyo Institute of Technology, 2-12-1 Ookayama, Meguro, Tokyo 152-8550, Japan. [9] Present address: Bayerisches Geoinstitut, University of Bayreuth, 95440 Bayreuth, Germany. ✉email: Keisuke.Nishida@uni-bayreuth.de

The Mars is the best studied planet except our own, but its interior remains largely unknown because seismological observations have not been performed yet. Geodesy studies indicated that the Mars has a liquid core[1,2]. The InSight mission is now in progress and has already revealed that Mars is seismically active[3]. Seismological observations are expected to reveal whether the core is fully molten, partially molten, or solid, and to constrain the composition of the core.

The Martian core has been thought to consist of Fe–S alloy because Mars is a volatile-rich planet[4] and Martian meteorites are depleted in chalcophile elements[5]. For the interpretation of seismic wave speeds, the knowledge of $V_P$ of liquid Fe–S alloy as functions of pressure, temperature, and sulfur concentration is necessary. However, previous measurements were made only up to 8 GPa[6,7] with high precision in a multi-anvil press, much lower than the likely pressure range of the Martian core (~20 to ~40 GPa)[2,8].

In this study, we determined the $V_P$ of liquid Fe, $Fe_{80}S_{20}$ and $Fe_{57}S_{43}$ up to 20 GPa, likely corresponding to the pressure ($P$) at the uppermost core of Mars[2,8]. The $P$–$V_P$ data obtained are extrapolated to conditions at the center of the Mars (~40 GPa)[2,8] based on thermodynamical equation of state. We found that sulfur have little effect on the $V_P$ of liquid iron in the Mars' whole core pressure range as opposite to the case for the core of the Moon (~5 GPa)[6]. It is therefore difficult to estimate sulfur content of Martian core based only on velocity even though the Martian core is molten and its seismic velocity will be determined. Alternatively, if the seismic velocity deviates from the values we obtained here, it indicates the presence of considerable amounts of impurity elements other than sulfur.

## Results and discussion

**Sound velocity of liquid Fe–S.** We measured the $V_P$ of liquid Fe, $Fe_{80}S_{20}$ and $Fe_{57}S_{43}$ based on ultrasonic pulse-echo method in a Kawai-type multi-anvil press up to 20 GPa at the SPring-8 and KEK-PF synchrotron radiation facilities in Japan (Fig. 1, Supplementary Fig. 1 and Supplementary Table 1). Temperature effect on the $V_P$ is found to be smaller than experimental uncertainty and is regarded negligible in this study. The velocity decreases with increasing sulfur concentration at low pressure range (Fig. 2a). Nevertheless, pressure effect is larger for Fe–S alloys than for pure Fe, and the $V_P$ of liquid $Fe_{80}S_{20}$ approaches that of liquid Fe around 10 GPa. Then, the velocity/pressure slope ($dV_P/dP$) for $Fe_{80}S_{20}$ diminishes and becomes similar to that for pure Fe at higher pressures. The $V_P$ of liquid $Fe_{57}S_{43}$ also approaches those of liquids Fe and $Fe_{80}S_{20}$ above 20 GPa (Supplementary Note 1, Supplementary Figs. 2 and 3).

The high $dV_P/dP$ in liquid Fe–S observed below 10 GPa is not found in liquid Fe. Solid FeS-V is known to exhibit anomalous volume contraction, which is attributed to high-spin to low-spin transition below 13 GPa[8] (Fig. 2b). Liquid FeS is also expected to undergo the spin crossover and indeed exhibits small bulk modulus $K_0$ at 1 bar and large pressure derivative $K'$ [9,10], leading to high $dV_P/dP$. It is possible that liquid $Fe_{80}S_{20}$ is an inhomogeneous mixture of a portion with the Fe-like structure and that with the FeS-like one, as is observed in the Se–Te system[11,12] (Supplementary Note 2). This is supported by a sigmoidal shape of the liquidus curve in the Fe–FeS system observed from 1 bar to 10 GPa[13,14], which indicates the presence of metastable two-liquid solvus at intermediate compositions (Supplementary Fig. 4a). Above 10 GPa, the sigmoidal liquidus curvature disappears, suggesting a nearly ideal solution[14] (Supplementary Fig. 4b); the structure of liquid $Fe_{80}S_{20}$ becomes homogeneous by 10 GPa due to spin transition in the FeS-like

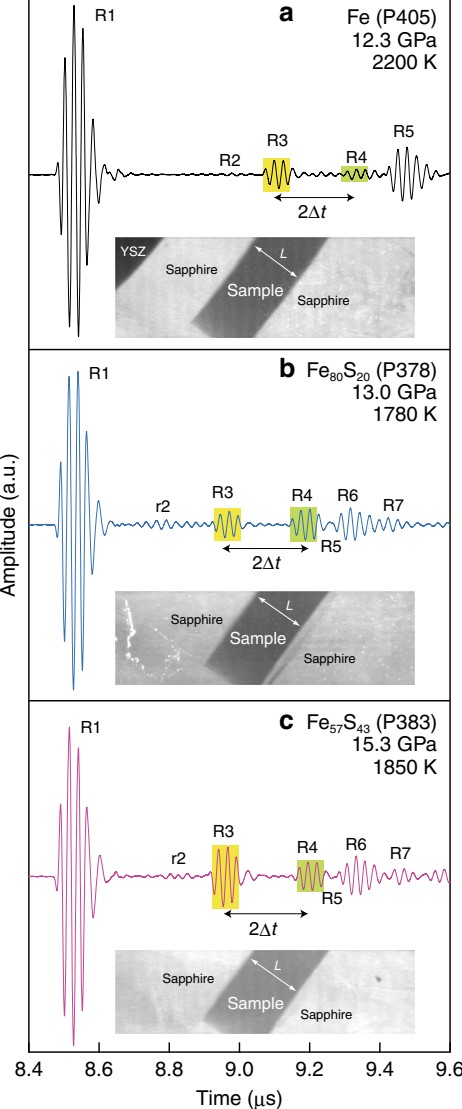

**Fig. 1 Examples of ultrasonic waveform and X-ray radiographic image of a fully molten sample.** (**a**) Fe, (**b**) $Fe_{80}S_{20}$, and (**c**), $Fe_{57}S_{43}$. R1–R7 represent echo signals by 3-cycle sine-wave burst with a center frequency of 40 MHz from the following boundaries; R1 = anvil/buffer-rod, R2 = YSZ/sapphire, r2 = $ZrO_2/Al_2O_3$ (surroundings), R3 = fronting sapphire/sample, R4 = sample/backing sapphire, R5 = sapphire/pressure marker (**a**) sapphire/c-BN (**b**, **c**), R6 = c-BN/pressure marker, R7 = pressure marker/MgO. See more detail in Supplementary Fig. 7. $L$ in X-ray radiographic image represents sample length; 517.7(5) μm for Fe, 507.7(1) μm for $Fe_{80}S_{20}$ and 541.7(6) μm for $Fe_{57}S_{43}$. 2$\Delta t$ in ultrasonic waveform represents two-way travel time in the sample; 221.1 ns for Fe, 217.0 ns for $Fe_{80}S_{20}$ and 241.7 ns for $Fe_{57}S_{43}$. Sound velocity ($V_P$) can be obtained as $L/\Delta t$.

portion. This interpretation does not contradict earlier structural studies using X-rays[15–18].

**Implications for Martian core.** In order to extrapolate the present $P$–$V_P$ data to >40 GPa corresponding to conditions at the center of the Mars[2,8], we fit adiabatic, third-order finite strain, Birch-Murnaghan equation of state to the data, assuming no temperature dependence (see Methods). Considering the effect of spin crossover in the FeS-like portion in liquid and resulting structural homogenization, only data collected at ≥10 GPa and ≥6.6 GPa were used for fitting for liquids $Fe_{80}S_{20}$ and $Fe_{57}S_{43}$, respectively. The extrapolations show that the velocities of the

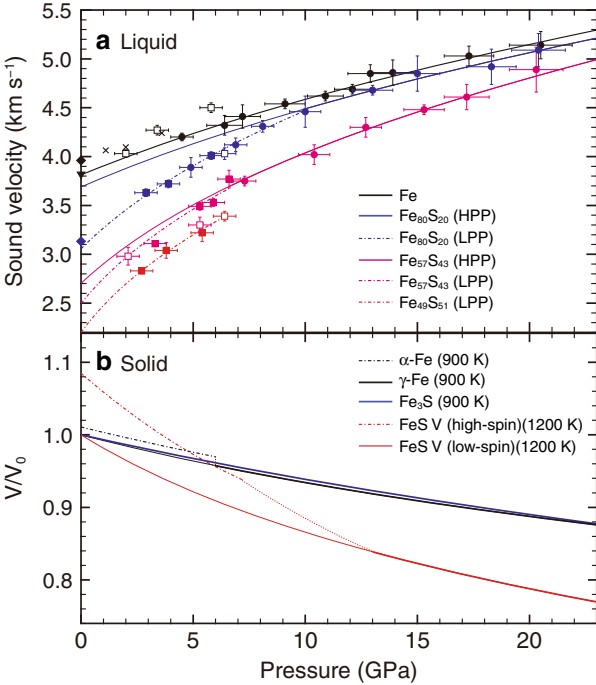

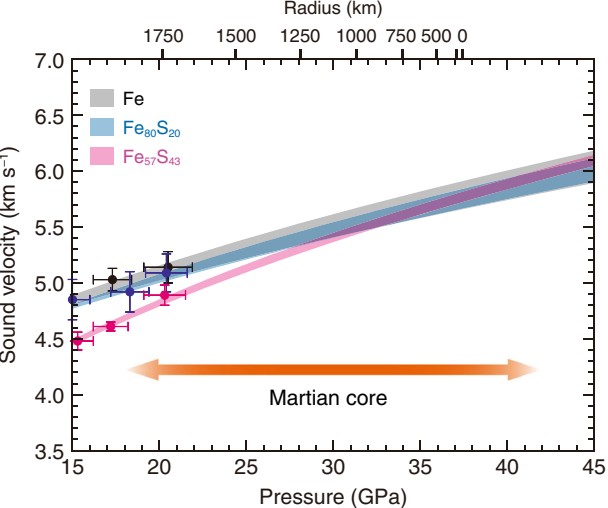

**Fig. 3 Pressure effect on $V_P$ in liquid Fe–S under Martian core conditions.** Hatched areas denote the ranges of $V_P$ in liquid Fe (gray), $Fe_{80}S_{20}$ (blue) and $Fe_{57}S_{43}$ (pink) extrapolated with and without using shock-wave data. Radius is based on ref. [2].

**Fig. 2 Sound velocity in liquid Fe and Fe–S alloys and compression curve for solids. a** $V_P$ data for liquid Fe (black), $Fe_{80}S_{20}$ (blue), $Fe_{57}S_{43}$ (pink) and $Fe_{49}S_{51}$ (red). Open and solid squares[6] and crosses[7] denote ultrasonic $V_P$ from previous studies. Solid reverse triangle[38] and diamonds[35,39] show 1-bar data. Solid curves denote the best fits for liquid high-pressure phase (HPP) without using shock data. Dash-dotted curves are for low-pressure phase (LPP). **b** Compression curves of solid α-Fe[40], γ-Fe[41], $Fe_3S$[42], and FeS V[8].

liquid Fe–S alloys could be faster than or at least are nearly identical to that of liquid pure Fe under deep Martian core conditions (Fig. 3). Indeed, our data are consistent with previous first-principles molecular dynamics simulations[19–21] when extrapolated to Earth's core pressures (Supplementary Fig. 5).

Interestingly the effect of sulfur on the $V_P$ of liquid Fe is found to be quite minor, just in the Martian core pressure range (Figs. 3 and 4). The model[4,5,22–24] suggests the bulk Martian core includes 16–36 at% S (Supplementary Note 3). The velocity of liquid $Fe_{80}S_{20}$ overlaps with that of liquid pure Fe within uncertainty in the whole Martian core pressure range between ~20 and ~40 GPa. A reduction in $V_P$ is only less than 0.3% per atomic % S at maximum (between $Fe_{80}S_{20}$ and $Fe_{57}S_{43}$ at 20 GPa) (Fig. 4), unlike the case for the cores of the Earth (>135 GPa) and the Moon (~5 GPa)[6]. The Martian core model[2] predicted its seismic velocity that is very close to that of liquid Fe–S found in this study for Mars' core conditions, suggesting that the travel time curve[25] calculated for their model[2] may be observed. However, considering uncertainty in velocity determinations for the Martian core in the near future, the velocity will not tell us its sulfur content. On the other hand, if the seismic velocity is different from the values we obtained here (Fig. 4), it precludes the Fe–FeS binary liquid for Mars' core and alternatively suggests that it contains other impurity element(s). For example, it has been reported[26,27] that the $V_P$ of liquid Fe is enhanced by the incorporation of carbon and silicon at 20–40 GPa. They will therefore be alternative candidates for the light element in the Mars' core if higher velocity is obtained. The effects of the other possible light elements such as hydrogen and oxygen remain to be explored at pressures relevant to the Martian core.

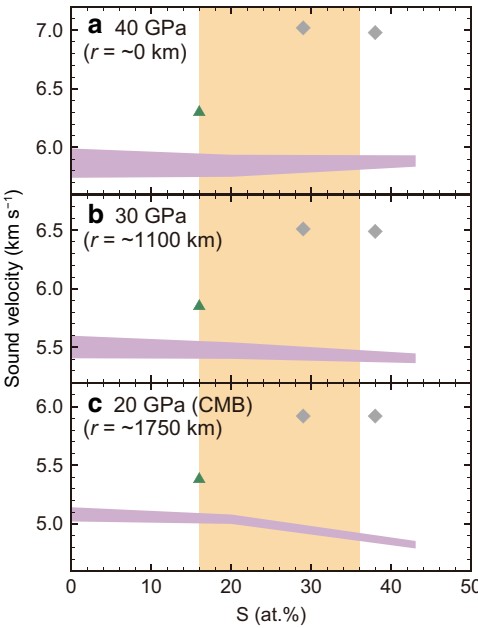

**Fig. 4 Sound velocity of liquid Fe–S alloy as a function of sulfur content under Martian core conditions.** (**a**) 40 GPa (Mars' center); (**b**) 30 GPa; (**c**) 20 GPa (Martian core-mantle boundary). Purple bands show the present estimates. Orange represents the sulfur content expected in the Martian core[4,5,22–24]. Green solid triangles denote $V_P$ measured for liquid $Fe_{84}C_{16}$[26]. Gray diamonds denote $V_P$ for $Fe_{61}Ni_{10}Si_{29}$ and $Fe_{52}Ni_{10}Si_{38}$ calculated from ref. [27]. The radius, $r$, is based on ref. [2].

## Methods

**Velocity measurements at high *P–T*.** High-pressure and -temperature (*P–T*) experiments were conducted using the Kawai-type multi-anvil apparatus MAX-III and SPEED-Mk.II at the beamline AR-NE7A, KEK-PF and at the beamline BL04B1, SPring-8, respectively. The starting materials were pure Fe powder or a powder mixture of Fe and FeS.

Cell assemblies are shown in Supplementary Fig. 6. We used a cylindrical TiC-$Al_2O_3$ composite heater. A semi-sintered $Al_2O_3$ sleeve was used as a thermal insulator. We adopted single-crystal sapphire as a buffer rod and a backing plate with a BN flat-bottomed cylindrical container for liquid Fe–S (type-1 cell assembly). Since chemical reaction between BN and liquid pure Fe was found in a preliminary experiment (the reaction with liquid Fe–S was limited), we also

employed type-2 cell assembly, in which sample was surrounded by a single-crystal sapphire sleeve, sintered polycrystalline yttria-stabilized zirconia (YSZ) rod and single-crystal sapphire plate lids (ultrasonic reflectors). Pyrophyllite gaskets were V-grooved (110°) and baked at 973 K for 30 min. It has been demonstrated[28] that difference in pressure between a sample fully surrounded by alumina and a pressure marker disappeared above 1073 K. Indeed, the present data for liquid $Fe_{80}S_{20}$ using the type-1 and type-2 cell assemblies are consistent with each other.

We used tungsten carbide (WC) anvils with 22 mm edge lengths as second-stage anvils in MAX-III, and those with 26 and 27 mm edge length in SPEED-Mk. II. The 22 and 26 mm WC anvils were made of Tungaloy F grade and employed for experiments at <15 GPa. The 27 mm ones were made of Fujilloy TF05 grade and used for >16 GPa experiments. The truncated edge length of the anvil face to a pressure medium was 5 mm.

Sound velocity was obtained by ultrasonic pulse-echo overlap method (Fig. 1 and Supplementary Fig. 7). Longitudinal-wave signals were generated and received by the 36° Y-cut $LiNbO_3$ transducer. To transfer high amplitude waveform to a sample, non-coated $LiNbO_3$ crystal with diameter of 4 or 5 mm was mounted onto the opposite corner of WC anvils with 22 mm and 26/27 mm edge length, respectively. A conducting epoxy layer was placed as an electrode on the $LiNbO_3$ crystal and worked also as a backing material to reduce ringing noise. A three-cycle sine wave burst with frequency of 35–60 MHz was used as an input electrical signal. The echo signal was attenuated and distorted to a large extent when those with >50 MHz were transferred. These phenomena were remarkable, in particular when the 26/27 mm anvils were used. Therefore, the frequency effect on travel time cannot be evaluated quantitatively but is considered to be much smaller than the effects of attenuation and distortion. In contrast, echoes sometimes overlapped with each other when the 35 MHz sine wave burst was transferred. Therefore, the velocities determined at 40 or 45 MHz were employed in this study. Two-way travel time in a sample was obtained by cross correlation function between echoes from the buffer-rod/sample and sample/backing. Sample length was estimated by an X-ray radiographic image analysis. The error in measured sound velocity was mainly derived from the uncertainty in the sample length.

Energy-dispersive X-ray diffraction (XRD) measurements were performed to obtain sample pressures. We used white X-rays with a Ge-solid state detector at the fixed diffraction angle of 6.0°. Pressure and temperature were determined simultaneously so that a couple of pressure standards (h-BN and MgO or NaCl and MgO) give identical pressure[29–31]. A thermocouple was not employed because it causes unwanted deformation of the sample and reduces the accuracy of measurements and could be a source of chemical contamination.

We checked the difference in temperature among pressure marker and several points of the sample position using the type-1 cell assembly, in which sample was replaced by a couple of pressure standards, NaCl and MgO. Temperatures in a sample chamber ranged from about 30 K lower to 60 K higher than that at the usual pressure marker position at ~15 GPa and ~1740 K.

Data acquisition flow is given in Supplementary Fig. 8. Upon each temperature increase by 100–200 K, we first obtained radiographic image and ultrasonic echo signals simultaneously, then XRD data, and collected the radiographic image and ultrasonic signals again. The molten state of a sample was indicated by XRD diffuse signals that are characteristic of liquid. We also employed changes in the radiographic image and ultrasonic signals to identify melting. We kept the molten state of a sample for 1–4 h in total, with data collection time of 7–15 min at each temperature. See Supplementary Fig. 9 for comparison between melting temperatures recognized in the present experiments and those reported in earlier studies.

**Chemical analyses of recovered samples**. The textures and chemical compositions of recovered samples were examined using a field-emission-type EPMA-WDS (JEOL, JXA-8530F) (Supplementary Tables 1, 2). We performed quantitative analyses with 12 kV acceleration voltage and 12 nA beam current using LIF (for Fe), PETH (for S) and LDE1 (for O) crystals, employing analytical standards of pure iron, chalcopyrite and hematite. ZAF correction was applied. We used a defocused beam with a diameter of 15 μm for the $Fe_{80}S_{20}$ and $Fe_{57}S_{43}$ samples, considering the presence of holes and cracks. For the Fe sample, a focused beam (1 μm) was employed. For samples that suffered contamination by Ti, C and Al, their concentrations were obtained with a defocused beam (10 μm) using PETJ (for Ti), LDE2H (for C) and TAP (Al) crystals employing analytical standards of $TiO_2$, $Fe_3C$ and corundum. ZAF correction was also applied.

Supplementary Fig. 10 shows typical back scattered electron images of recovered samples. In experiments with the type-1 assembly, we found no or little dendritic BN in the quenched liquids $Fe_{80}S_{20}$ and $Fe_{57}S_{43}$. Additionally, while small dendritic oxide crystals were sometimes formed near the buffer rod and the capsule (Supplementary Fig. 10e, f), it should have little affected the $V_P$ measurements because such oxide crystals were very minor and located far from the center of the sample. When adsorbed water was not carefully removed, a large amount of FeO was observed near the buffer-rod and the capsule. Dendritic B–N–O or B–N–S crystals were also found in the Fe–S matrix uniformly in these experiments. We excluded such experiments from this study.

Contamination by a minor amount of oxygen could have occurred during sample preparation for EPMA analysis, and therefore the oxygen content in a liquid sample may have been smaller. In addition, several recovered samples

showed contamination from a $TiC-Al_2O_3$ heater. Radiographic images indicated that the contamination occurred during cooling (i.e. after measuring liquid) in runs #P393 and #P404. On the other hand, it is uncertain when it happened in runs #M2413 and #M2415. The quantitative EPMA analyses of the Ti, C, Al, and O contents in quenched liquids are given in Supplementary Table 2 for these two runs. Indeed, the $P–V_P$ curve changes very little with these two data points.

**Data extrapolation**. $V_P$ of a liquid corresponds to bulk sound velocity and is written as

$$V_P(P, T) = \sqrt{\frac{K_S(P, T)}{\rho(P, T)}} \quad (1)$$

in which $K_S$ is adiabatic bulk modulus. The present measurements show a small temperature effect on $V_P$, and thus;

$$V_P(P, T) \approx V_P(P, T_S) = \sqrt{\frac{K_S(P, T_S)}{\rho(P, T_S)}} \quad (2)$$

with $T_S$ of temperature on a reference isentrope. $\rho(P, T_S)$ and $K_S(P, T_S)$ can be expressed by the adiabatic third-order Birch–Murnaghan EoS as

$$P = 1.5K_{S0}\left(\frac{\rho}{\rho_0}\right)^{\frac{5}{3}}\left[\left(\frac{\rho}{\rho_0}\right)^{\frac{2}{3}} - 1\right]\left[1 + 0.75\left(K_S' - 4\right)\left\{\left(\frac{\rho}{\rho_0}\right)^{\frac{2}{3}} - 1\right\}\right], \quad (3)$$

$$K_S = K_{S0}\left(\frac{\rho}{\rho_0}\right)^{\frac{5}{3}}\left[1 + 0.5(3K_S' - 5)\left\{\left(\frac{\rho}{\rho_0}\right)^{\frac{2}{3}} - 1\right\} + \frac{27}{8}\left(K_S' - 4\right)\left\{\left(\frac{\rho}{\rho_0}\right)^{\frac{2}{3}} - 1\right\}^2\right] \quad (4)$$

in which $K_S'$ is the pressure derivative of $K_S$ and subscript zero indicates values at ambient pressure. We fit Eqs. (2–4) to the present $P–V_P$ data (Fig. 2 and Supplementary Fig. 1). The $P–V_P$ fitting curve is practically independent on the choice of $\rho_0$ (Supplementary Fig. 11), while there is a trade-off between $\rho_0$ and the combination of $K_{S0}$ and $K_S'$. In other words, when $\rho_0$ is unknown, it is difficult to obtain $\rho$ and $K_S$ precisely but is possible to extrapolate $V_P$ to higher pressure.

In the cases of $Fe_{80}S_{20}$ and $Fe_{57}S_{43}$, only the data collected ≥ 10.0 GPa and ≥6.6 GPa were fitted, respectively, considering the effect of spin crossover in the FeS-like portion in liquids. We also performed fitting to our $P–V_P$ data together with previous shock data on Fe[32], $Fe_{80}S_{20}$[33], and $Fe_{57}S_{43}$ (ref. [34] for FeS). Fitting parameters obtained by weighted least-squared fitting are summarized in Supplementary Table 3. The $K_{S0}$ and $K_S'$ values for liquid Fe obtained together with shock-wave data agree with those reported in earlier studies[35,36]. The reference density for liquid Fe-S should be the one for the high-pressure structure where spin transition is complete. Here we chose the $\rho_0$ values for liquids $Fe_{80}S_{20}$ and $Fe_{57}S_{43}$, which explain the $P–\rho$ data obtained by shock experiments[33]. Such $\rho_0$ for liquid $Fe_{80}S_{20}$ is, however, not consistent with the other $P–\rho$ relations by theory[21] and experiments[37]. Therefore, the $\rho_0$, $K_{S0}$ and $K_S'$ for liquids $Fe_{80}S_{20}$ and $Fe_{57}S_{43}$ listed in Supplementary Table 3 are not certain.

As shown in Supplementary Fig. 1, the fitting curves reproduce the experimental data well. There is no significant difference between those with and without considering the shock-wave data in the present experimental pressure range. In the Earth's core pressure range (>135 GPa), the extrapolated $V_P$ of liquids Fe and $Fe_{80}S_{20}$ without shock data is slightly higher than that considering them but matches the results by first-principles molecular dynamics simulations[19–21] (Supplementary Fig. 3).

The present $P–T$ conditions of measuring the velocity of liquid $Fe_{80}S_{20}$ are illustrated in Supplementary Fig. 12. While temperature ranged from 1700 to 2240 K, we obtain a single $P–V_P$ relation (Fig. 2), indicating that $V_P$ is insensitive to temperature. Our data may be applicable to the Martian core without correcting for a temperature difference.

## Data availability
The data supporting the main findings of this study are available in the paper and its Supplementary Information. Any additional data can be available from the corresponding author upon reasonable request.

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

## Acknowledgements

The authors acknowledge K. Ichimura, H. Yoshida and K. Yonemitsu for assisting EPMA analyses. We also thank Y. Tange, S. Kamada, H. Tobe, R. Abe, S. Kobayashi, I. Yamada, Y. Shimoyama, and K. Watanabe for their advice and technical support. Comments from anonymous reviewers helped to improve the manuscript. This work was supported by JSPS KAKENHI (grant no. 12J07930, 26800231, 17K14379, and 16H06285). The synchrotron radiation experiments were performed at the BL04B1 beamline at the SPring-8 facility (proposal no. 2013A1508, 2013B1174, 2014A1146, 2016A1235, 2017A1255, and 2017B1270) and at the AR-NE7A beamline at the KEK PF-AR facility (proposal no. 2015G539 and 2017G634).

## Author contributions

K.N. organized the research project and carried out most of the experiments and data analyses with help by Y.S., H.T., A.S. and Y.H. All authors discussed experimental results and their geophysical implications. The manuscript was written by K.N., N.F., and K.H.

## Competing interests

The authors declare no competing interests.
