## [Peer Review File · Nature Communications]

Reviewers' Comments:

Reviewer #1:

Remarks to the Author:

The paper "Effect of sulfur on sound velocity of liquid iron under Martian core conditions" describes the results of high-pressure-high-temperature laboratory experiments on the velocity of seismic P-waves in liquid Fe-S mixtures with different Sulphur concentrations. Experiments are carried out at pressures relevant for the martian core-mantle boundary and results extrapolated to the pressure at the center of Mars. The obtained velocities can be used to test the hypothesis that the martian core is a Fe-S mixture. As NASA's InSight mission is currently recording seismic events on Mars, it is timely and necessary to have this kind of result at hand when it comes to the interpretation of relevant seismic data.

I should first make clear that I am not a specialist for the kind of experiment described in the paper, but a seismologist who is mainly interested in the applicability of the results. I can therefore not really judge the quality of the experiments themselves and trust that another reviewer will do so.

It is rare, although highly desirable, to get a clear statement like the authors make: if the velocities measured in the martian core are outside the experimentally determined range, then a Fe-S composition can be ruled out. It would be great to have more such results for other types of composition (as a number of light elements have been proposed to be present in the martian core). This is not intended as criticism, since the authors can obviously not do everything at once.

Although the determination of the martian core radius is one of the main goals of InSight, this does not necessarily imply that the P-wave velocity directly underneath the CMB will be available at the same time. The core radius may be determined from the extent of the core shadow or from travel times of seismic phases like PcP and ScS - such results will not provide a look into the uppermost core. Also, since the core likely has lower P wave velocity as the lowermost mantle, it will not be possible to determine P wave velocity from the slope of travel time curves, as there will be a core shadow (and hence no travel time curve for the uppermost part of the core). This limits the applicability of velocity as criterion. Although it is not the purpose of this paper to describe an appropriate seismic experiment, it would certainly be strengthened if the authors could include some comments on the type of seismic phases needed. I do not consider this mandatory.

A more important point, which I do consider mandatory, concerns the interpretation of seismic velocities measured by the authors. The figure showing seismic waveforms and phase identifications is nice and clear, I was however missing a schematic view that explains the different recorded phases (termed R1, R2, R3 and so on) in terms of the propagation paths through the sample. I suggest the authors include a diagram showing the sample geometry, the locations of sources and sensors, and the ray paths associated to the different phases indicated in the current figure. Also the figure contains labels ("L" and "2T") which are not explained in the caption nor in the text, such that it is unclear what they are and how they are related. This point is of utmost importance, since the velocity result depends critically on the correct identification of phases, and the associated propagation paths.

The omission of error bars in a figure in the main text is, in my opinion, not acceptable. Not all readers will download and inspect the supplementary material upon first read, but they should nevertheless find the uncertainty information. I did not have the impression that the figures containing error bars in the supplement are unreadable. When I see a figure where the caption states that error bars are omitted for clarity, I immediately assume that errors are so big that fitting a curve is meaningless.

An information I was also missing in the manuscript is the acoustic wavelength used for sounding the samples, and how it relates to sample size and path length. Please add a few numbers in this

regard.

In some cases, fitted curves shown in the supplement do not fit well to some of the data points, but are very close to others. May be the authors decided to fit only part of the data, but this needs to be indicated (and justified), and otherwise it needs to be discussed why some fits look poor.

For the center of Mars, the authors could only extrapolate their experimental results. Although I understand that this might be due to technical limitations of the apparatus (which needs to be indicated), I recommend that the authors discuss if it can be expected that the assumed mixtures remain liquid at such pressures and temperatures, and what the liquidus and solidus conditions are expected to be. This must be seen in the light of the open question whether or not Mars has a solid core.

In general, seismologists prefer thinking in terms of depth rather than of pressure. The uncertainty of the martian core radius, based on current knowledge, is several hundred km (may be up to +/- 500 km, when pessimistic). The authors should indicate what core radius (or range) is associated to the pressures used in their experiments.

The authors give, at the end of the paper, the names of some beam types used to "x-ray" their samples. To me, the given designations do not mean much. I would find it helpful if the names of the facilities and institutions running the facility, and the location of the facilities, could be added.

Reviewer #2:

Remarks to the Author:

This paper by Nishida et al. on Effect of sulfur on sound velocity of liquid iron under Martian core conditions reported the sound velocity of liquid Fe and Fe-S alloys using the ultrasonic method combined with large volume press and synchrotron X-ray up to 20 GPa and 2320 K, corresponding to the Martian core conditions. The velocity and density data of liquid Fe core alloys with a reasonable accuracy are required to evaluate such planetary seismological observations. However, the pressure range of such data has been stacked at below 10 GPa in last decades. The authors broke the situation and succeeded to measure the velocity of liquid Fe-S alloys to 20 GPa, that is double pressure range previously reported and reaches to Martian core conditions. Based on the present results, the authors model the seismic wave speeds through the Martian core as a function of sulfur content and found that the seismic profile of Martian core is insensitive to its sulfur content, which can be useful to evaluate often-used sulfur-rich Martian core model with the seismological data from the ongoing NASA's Insight mission.

The manuscript is well written, and the presented materials adequately illustrate the methodology and results. The authors' achievement stands on the technical development as well as very-careful experiments by the authors including the steady evaluations of the velocity data and recovered samples. The sound velocity data presented here therefore appear to be of really-good quality. The seismic observations are one of the main projects of the Insight mission and the Martian core composition is strongly related to the building blocks and core formation process, such that the present results will be a key role in the mission and be spread to a number of fields such as cosmochemistry, planetary sciences, astronomy sciences, and other communities involved in the planetary explore mission. The developed technique here will be also applied to the fundamental physics like disordered-material physics as well as to material-sciences related to metallic glass and steel making. Therefore, I recommend this paper to be published in Nature Communications after fixing several issues I commented below.

Major comments:

-The authors claim that the sound velocity of liquid Fe is insensitive to the sulfur content and the Fe-S liquid Martian core is ruled out if the Martian core velocity was deviated from that of pure Fe

(L40, L65-66, L113-115). This statement sounds funny for me. The deviation from pure Fe or Fe-S liquids cannot rule out the sulfur-rich Martian core because both Fe and S are not distinguishable from the velocity only as presented by the authors (L62-63). This statement can be established if the total Fe content (or light elements) in the Martian core is fixed from other evidence such as density and geochemical observations. Please carefully reconsider this statement.

-The authors claim the EoS parameters presented here are "not physical properties just fitting parameters" (L227). If so, the Birch-Murnaghan EoS applied in this study also loses its validity because the EoS formula is based on the potential energy change by the finite strain and its parameters are resulted from the thermodynamically derivatives of the energy. As the authors suggested, it can be hard by the only present data to investigate the physically-meaningful density for the high-pressure liquid Fe-S. However, it is possible to evaluate the obtained density or its temperature when using previously reported the density data of liquid Fe-S by the authors (Nishida et al. 2011AmeMine; 2013PCM; Terasaki et al. 2019JGR; Umemoto et al. 2014JGR) and/or by the others (Morard et al. 2013EPSL; Huang et al. 2018JGR).

Minor comments:

- "..., only data collected at ~13 GPa and higher..." (L99-100), how did the authors chose this value? Though the slope in Fig. S1 looks to change around ~10 GPa for Fe₈₀S₂₀ but at ~6 GPa, more quantitative evaluation for the critical pressure is required.

- In "Method". It is well known that it is difficult to keep the liquid state under high-pressure and - temperature for long time. The authors may carefully check the liquid during measurements based on XRD from the samples. The textural image of recovered samples of Fe-S can be evident of the liquid state during ultrasonic measurements. However, it is hard to judge the liquid state of pure metal such as Fe from recovered samples. I recommend the authors to show the XRD patterns for the judgment during experiments. In addition, please describe the time scale of the ultrasonic measurements for the present liquid samples, in other words, how long the authors kept the liquid under high pressure and temperature for the measurements.

-Please give the uncertainties for the fitting parameters in "Supplementary Table 3".

- "assuming g is constant" (L243). It is unclear what values for which alloys the authors assumed here. The Grüneisen parameter has the volume and composition dependency. Also, several values for the Grüneisen parameters among Fe-S alloys are listed in the referred papers. Please clarify which values used here.

-In Fig. S3, the symbols of "This study" and "I14" are hardly distinguished by my eyes. Please make clear difference among the symbols.

-In Fig. S6, i of Ni in the legend should not be subscript.

-Table S1. I am worry that the authors overestimate the oxygen contamination in the recovered sample. Usually, small amount of O can be detected even pure metal due to the surface oxidation of metallic samples during the sample preparation. Did you consider such an artificial oxygen in the EPMA analysis?

Reviewer #3:

Remarks to the Author:

This manuscript reports new experimental data on the sound velocity of liquid Fe and two liquid Fe-S alloys (Fe₈₀S₂₀ and Fe₅₇S₄₃) to 20 GPa. These data add to existing data at relatively low pressures, but demonstrate a different pressure dependence. The manuscript claims that if these data can be extrapolated to to Martian core pressures (20-40 GPa), then there would be little

effect of S on the velocity of liquid Fe. If this is the case, then future seismic data from planetary missions may not be used to constrain the S content in the core, but may suggest the presence of light elements other than S, depending on the actual seismic velocity observations.

These are challenging experiments and the data are important to constraining the composition of the liquid outer core of Mars. However, there are some issues with the experiments as well as the data analysis as detailed below that could result in large uncertainties with the extrapolation, hence weaken the major point of the paper. These need to be clarified and additional measurements may be required to validate the results of the paper.

1. I am concerned with the consistency of the data presented in this manuscript with some of the co-author's previous data published in Terasaki et al, 2019, Journal of Geophysical Research, doi: 10.1029/2019JE005936. This manuscript does not mention Terasaki et al. (2019) at all, but Terasaki et al. (2019) performed similar experiments on Fe-10Ni-17S and Fe-10Ni-30S up to 14 GPa, and showed a different pressure dependence of V_p for the S-richer composition than observed in this study. Although there is some difference in composition that samples in Terasaki et al. contained Ni, but as discussed Terasaki et al, Ni has little effect on the velocity. The Terasaki et al. 2019 results need to be compared in this paper, and the discrepancy needs to be discussed and clarified in the manuscript.
2. The sound velocity data for the Fe-S liquids were analyzed using Eqs (2-4) without the knowledge of density. The manuscript argued that although this would constrain the equation of state, but would affect the extrapolation for V_p . This needs to be demonstrated in more detail, for example different fittings with different assumed densities need to be performed to show the effect of the choice of density. It is potentially dangerous to leave one parameter unconstrained during the extrapolation (interpolation might be OK). Also, the calculation of adiabatic temperature (eq. (5)) requires the knowledge of density. The uncertainties in T_s due to the choice of density should also be discussed.
3. The sound velocity of liquid Fe was determined using the type 2 assembly, in which the sample was fully surrounded by single crystal sapphire. Because of the hardness of single crystal sapphire, it is questionable that the sample inside would achieve the same pressure as indicated by the pressure markers that were partially surrounded by soft materials. This would result in some overestimation of the experimental pressure in these experiments, and hence could at least partially explain the discrepancy in the sound velocity of liquid Fe between this study and those from references 6, 7, and 27, 28. This needs to be examined experimentally and discussed in the manuscript.
4. In Figure 4, it can be seen that there is clearly some effect of S on V_p at 20 GPa and 30 GPa (as large as a few percent). It is not clear to me why this much effect cannot be distinguished from future seismic observations as claimed in the paper. Some more quantitative discussion may help explain this.

Reply to Reviewer #1: (Reviewer's comment in blue italic, and our response in black)

The paper "Effect of sulfur on sound velocity of liquid iron under Martian core conditions" describes the results of high-pressure-high-temperature laboratory experiments on the velocity of seismic P-waves in liquid Fe-S mixtures with different Sulphur concentrations. Experiments are carried out at pressures relevant for the martian core-mantle boundary and results extrapolated to the pressure at the center of Mars. The obtained velocities can be used to test the hypothesis that the martian core is a Fe-S mixture. As NASA's InSight mission is currently recording seismic events on Mars, it is timely and necessary to have this kind of result at hand when it comes to the interpretation of relevant seismic data.

I should first make clear that I am not a specialist for the kind of experiment described in the paper, but a seismologist who is mainly interested in the applicability of the results. I can therefore not really judge the quality of the experiments themselves and trust that another reviewer will do so.

It is rare, although highly desirable, to get a clear statement like the authors make: if the velocities measured in the martian core are outside the experimentally determined range, then a Fe-S composition can be ruled out. It would be great to have more such results for other types of composition (as a number of light elements have been proposed to be present in the martian core). This is not intended as criticism, since the authors can obviously not do everything at once.

Although the determination of the martian core radius is one of the main goals of InSight, this does not necessarily imply that the P-wave velocity directly underneath the CMB will be available at the same time. The core radius may be determined from the extent of the core shadow or from travel times of seismic phases like PcP and ScS - such results will not provide a look into the uppermost core. Also, since the core likely has lower P wave velocity as the lowermost mantle, it will not be possible to determine P wave velocity from the slope of travel time curves, as there will be a core shadow (and hence no travel time curve for the uppermost part of the core). This limits the applicability of velocity as criterion. Although it is not the purpose of this paper to describe an appropriate seismic experiment, it would certainly be strengthened if the authors could include some comments on the type of seismic phases needed. I do not consider this mandatory.

As the Reviewer pointed out, it is possible that the core P-wave is slower than the S-wave in the lowermost mantle and therefore the uppermost core phase may not be observed. Nevertheless, the sound velocity of liquid Fe-S under Mars core conditions obtained in this study is very similar to that of the Rivoldini model in Helffrich (2017 PEPS), and the calculated travel time curve model could be a good reference. Indeed, the model suggests possible observations of the SKS and SKKS phases. In addition, if the inner core exists, PKiKP and SKiKS can be observed.

In response to this comment, we have added the following sentence in the main text (Line 114–117);

“The Martian core model² predicted its seismic velocity that is very close to that of liquid Fe-S found in this study for Mars’ core conditions, suggesting that the travel time curve²⁵ calculated for their model² may be observed.”

A more important point, which I do consider mandatory, concerns the interpretation of seismic velocities measured by the authors. The figure showing seismic mwaveforms and phase identifications is nice and clear, I was however missing a schematic view that explains the different recorded phases (termed R1, R2, R3 and so on) in terms of the propagation paths through the sample. I suggest the authors include a diagram showing the sample geometry, the locations of sources and sensors, and the ray paths associated to the different phases indicated in the current figure. Also the figure contains labels ("L" and "2T") which are not explained in the caption nor in the text, such that it is unclear what they are and how they are related. This point is of utmost importance, since the velocity result depends critically on the correct identification of phases, and the associated propagation paths.

Following this advice, we have added Supplementary Fig. 5 (see below), which shows the locations of sources and sensors and the ray paths.

Supplementary Figure 5 | Deployment diagram of cell assembly for ultrasonic (before compression), and examples of ultrasonic echoes and their ray paths. Red lines (R1–R10 or R1–R9) in the cell show reflections that correspond to echo signals R1–R9 in the waveform diagram. **a**, Type-1 cell at 13.0 GPa and 1780 K (Run# P378) for the measurements of liquid Fe₈₀S₂₀. **b**, Type-2 cell at 12.3 GPa and 2200 K (Run# P405) for liquid Fe. Some unidentified echoes may be attributed to multiple reflections. Note that the cell size changed upon compression. See Fig. 1 for an enlarged waveform diagram.

The omission of error bars in a figure in the main text is, in my opinion, not acceptable. Not all readers will download and inspect the supplementary material upon first read, but they should nevertheless find the uncertainty information. I did not have the impression that the figures containing error bars in the supplement are unreadable. When I see a figure where the caption states that error bars are omitted for clarity, I immediately assume that errors are so big that fitting a curve is meaningless.

Based on this suggestion, we now give error bars in Fig. 2.

An information I was also missing in the manuscript is the acoustic wavelength used for sounding the samples, and how it relates to sample size and path length. Please add a few numbers in this regard.

While they are mentioned in the Methods section, we added these pieces of information in the Fig. 1 caption (see below).

“Figure 1 | Examples of ultrasonic waveform and X-ray radiographic image of a fully molten sample. a, Fe, b, Fe₈₀S₂₀ and c, Fe₅₇S₄₃. R1–R7 represent echo signals by 3-cycle sine-wave burst with a center frequency of 40 MHz from the following boundaries; R1, anvil/buffer-rod; R2, YSZ/sapphire; r2, ZrO₂/Al₂O₃ (surroundings); R3, fronting sapphire/sample; R4, sample/backing sapphire; R5, sapphire/pressure marker (a), sapphire/c-BN (b, c); R6, c-BN/pressure marker; R7, pressure marker/MgO. See more detail in Supplementary Fig. 5. “L” in X-ray radiographic image represents sample length; 517.7(5) μm for Fe, 507.7(1) μm for Fe₈₀S₂₀ and 541.7(6) μm for Fe₅₇S₄₃. “2Δt” in ultrasonic waveform represents two-way travel time in the sample; 221.1 ns for Fe, 217.0 ns for Fe₈₀S₂₀ and 241.7 ns for Fe₅₇S₄₃. Sound velocity (V_P) can be obtained as L/Δt.”

In some cases, fitted curves shown in the supplement do not fit well to some of the data points, but are very close to others. May be the authors decided to fit only part of the data, but this needs to be indicated (and justified), and otherwise it needs to be discussed why some fits look poor.

This is because we did not use the low-pressure data where spin transition is incomplete. While we mention it in the main text, we have modified the caption of Supplementary Fig. 1 as follows;

“Supplementary Figure 1 | Sound velocities in liquid Fe and Fe-S alloys at high pressure. Solid circles show the present ultrasonic V_P data. Solid and dashed curves are obtained by fitting without and with using previous shock-wave data for Fe (ref. 35), Fe₈₀S₂₀ (ref. 36) and FeS (ref. 37), considering the effect of spin crossover in the FeS-like portion in liquid (see text). a, Liquid Fe. Open squares⁶ and crosses⁷ denote ultrasonic V_P from previous studies. Solid reverse triangle²⁶ and diamond²⁷ show 1-bar data. Dash-dotted curve is calculated from the EOS³⁸. b, Liquid Fe₈₀S₂₀. Open and solid squares represent ultrasonic V_P from ref. 6. Solid diamond denotes the ultrasonic V_P of liquid Fe_{79.3}Ni_{4.4}S_{16.3} at 1 bar²⁸. c, Liquid Fe₅₇S₄₃. Open and solid squares show the previous ultrasonic V_P data for liquid Fe₅₇S₄₃ (pink) and Fe₄₉S₅₁ (red), respectively⁶.”

For the center of Mars, the authors could only extrapolate their experimental results. Although I understand that this might be due to technical limitations of the apparatus (which needs to be indicated), I recommend that the authors discuss if it can be expected that the assumed mixtures remain liquid at such pressures and temperatures, and what the liquidus and solidus conditions are expected to be. This must be seen in the light of the open question whether or not Mars has a solid core.

First of all, while the Reviewer mentioned “assume mixtures”, we argue in this paper that a microscopic mixture of Fe-like and FeS-like portions becomes homogeneous above ~10 GPa, meaning that the Martian core could consist of a homogeneous Fe-S liquid.

On the basis of this comment, we examined whether the Mars has a solid core by

comparing the liquidus temperatures in the Fe-FeS system and possible temperature profiles in the Martian core. For pure Fe, the present experimental data are consistent with relatively low melting curves (see newly added Supplementary Fig. 7a below), and here we employ the one by Boehler et al. (1990). For the FeS end-member, the melting curve exhibits a steep dT/dP slope at low pressures (Supplementary Fig. 7b). At higher pressure range, the one reported by Boehler et al. (1992) is consistent with both our data obtained around 20 GPa and shock-compression data, but does not agree with the low pressure experiments. It is therefore likely that the melting curve of FeS changes its dT/dP slope around 5 GPa (green curve in Supplementary Fig. 7b). Indeed, it could be because of spin crossover in liquid FeS, which is suggested from the present velocity measurements.

The present experiments demonstrated that the liquidus temperature of $\text{Fe}_{80}\text{S}_{20}$ increases linearly with pressure with the maximum at 18 GPa where Fe_3S is formed as an intermediate compound and the system changes into Fe- Fe_3S eutectic (Supplementary Fig. 7c). The liquidus curve of $\text{Fe}_{80}\text{S}_{20}$ is extrapolated to higher pressures by using a eutectic temperature at 36.8 GPa where $\text{Fe}_{80}\text{S}_{20}$ is a eutectic liquid composition (Stewart et al., 2007). The liquidus temperature of $\text{Fe}_{57}\text{S}_{43}$ decreases to 1.5 GPa where $\text{Fe}_{57}\text{S}_{43}$ is a eutectic composition (Usselman, 1975) and then increases rapidly to 5 GPa, likely due to the effect of spin crossover as we argued for the FeS end-member above. Subsequently the slope changes, and the liquidus curve of $\text{Fe}_{57}\text{S}_{43}$ becomes parallel to that of FeS (Supplementary Fig. 7d).

Supplementary Figure 7 | Liquidus curves in the Fe-FeS system. Open symbols denote completely molten state found in the present experiments, while solid symbols show solid or partially molten state. **a**, Fe. Green⁵⁷, red⁵⁸, black⁴⁶ and blue⁴⁷ curves from previous experiments. **b**, FeS. Yellow⁵⁰, gray⁵⁹, and red⁴⁸ curves were obtained for FeS and blue⁴⁹ for pyrrhotite. Spin crossover in liquid FeS likely changes the slope (green curve), which explains both our data and earlier results obtained at higher pressures. **c**, $\text{Fe}_{80}\text{S}_{20}$. Blue line shows the liquidus temperature estimated in this study. The results by Stewart *et al.*⁵² is given by red line. **d**, $\text{Fe}_{57}\text{S}_{43}$. Pink line indicates the liquidus temperature

determined in this study.

These liquidus curves under the Martian core pressure range are illustrated in newly added Supplementary Fig. 10 (see below). The estimated Mars' core temperatures are given by green and pink bands^{2,54}. It indicates that if the core composition is between $\text{Fe}_{80}\text{S}_{20}$ and $\text{Fe}_{57}\text{S}_{43}$, the core should be fully molten (no solid core), while solid core could exist when the Martian core composition is between Fe and $\text{Fe}_{80}\text{S}_{20}$ or between $\text{Fe}_{57}\text{S}_{43}$ and FeS.

Supplementary Figure 10 | P - T for present liquid $\text{Fe}_{80}\text{S}_{20}$ measurements in comparison to proposed Martian core conditions. Blue solid circles denote conditions for liquid $\text{Fe}_{80}\text{S}_{20}$ measurements in this study. Pink and green bands show Mars' core temperature models, ref. 2 and ref. 54, respectively. Liquidus curves are from Supplementary Fig. 7.

In response to this comment, we added a new section “**Liquidus and solidus temperature in the Fe-FeS system**” and these two new figures (Supplementary Figs. 7 & 10) to the Supplementary information.

In general, seismologists prefer thinking in terms of depth rather than of pressure. The uncertainty of the martian core radius, based on current knowledge, is several hundred km (may be up to +/- 500 km, when pessimistic). The authors should indicate what core radius (or range) is associated to the pressures used in their experiments.

Following this advice, radius is given, in addition to pressure, in Figs. 3 and 4 based on the depth-pressure model by Rivoldini et al. (2011).

The authors give, at the end of the paper, the names of some beam types used to "x-ray" their samples. To me, the given designations do not mean much. I would find it helpful if the names of the facilities and institutions running the facility, and the location of the facilities, could be added.

On the basis of this comment, we have added the following statement in Results (Line 70–73);

“We measured the V_P of liquid Fe, $Fe_{80}S_{20}$ and $Fe_{57}S_{43}$ based on ultrasonic pulse-echo method in a Kawai-type multi-anvil press up to 20 GPa at the SPring-8 and KEK-PF synchrotron radiation facilities in Japan (Fig. 1, Supplementary Fig. 1 and Supplementary Table 1).”

Reply to reviewer #2: (Reviewer's comment in blue italic, and our response in black)

This paper by Nishida et al. on Effect of sulfur on sound velocity of liquid iron under Martian core conditions reported the sound velocity of liquid Fe and Fe-S alloys using the ultrasonic method combined with large volume press and synchrotron X-ray up to 20 GPa and 2320 K, corresponding to the Martian core conditions. The velocity and density data of liquid Fe core alloys with a reasonable accuracy are required to evaluate such planetary seismological observations. However, the pressure range of such data has been stacked at below 10 GPa in last decades. The authors broke the situation and succeeded to measure the velocity of liquid Fe-S alloys to 20 GPa, that is double pressure range previously reported and reaches to Martin core conditions. Based on the present results, the authors model the seismic wave speeds through the Martin core as a function of sulfur content and found that the seismic profile of Martian core is insensitive to its sulfur content, which can be useful to evaluate often-used sulfur-rich Martian core model with the seismological data from the ongoing NASA’s Insight mission.

The manuscript is well written, and the presented materials adequately illustrate the methodology and results. The authors’ achievement stands on the technical development as well as very-careful experiments by the authors including the steady evaluations of the velocity data and recovered samples. The sound velocity data presented here therefore appear to be of really-good quality. The seismic observations are one of the main projects of the Insight mission and the Martian core composition is strongly related to the building blocks and core formation process, such that the present results will be a key role in the mission and be spread to a number of fields such as cosmochemistry, planetary sciences, astronomy sciences, and other communities involved in the planetary explore mission. The developed technique here will be also applied to the fundamental physics like disordered-material physics as well as to material-sciences related to metallic glass and steel making. Therefore, I recommend this paper to be published in Nature Communications after fixing several issues I commented below.

Major comments:

-The authors claim that the sound velocity of liquid Fe is insensitive to the sulfur content and the Fe-S liquid Martian core is ruled out if the Martian core velocity was deviated from that of pure Fe (L40, L65-66, L113-115). This statement sounds funny for me. The deviation from pure Fe or Fe-S liquids cannot rule out the sulfur-rich Martian core because both Fe and S are not distinguishable from the velocity only as presented by the authors (L62-63). This statement can be established if the total Fe

content (or light elements) in the Martian core is fixed from other evidence such as density and geochemical observations. Please carefully reconsider this statement.

We agree. Following this advice, we have written the relevant statements as follows;

“The comparison of seismic wave speeds of Fe-S liquids with future observations will therefore tell whether the Martian core is molten and contains impurity elements other than sulfur.” (Line 38–41)

“Alternatively, if the seismic velocity deviates from the values we obtained here, it indicates the presence of considerable amounts of impurity elements other than sulfur.” (Line 65–67)

“On the other hand, if the seismic velocity is different from the values we obtained here (Fig. 4), it precludes the Fe-FeS binary liquid for Mars’ core and alternatively suggests that it contains other impurity element(s).” (Line 118–121)

-The authors claim the EoS parameters presented here are “not physical properties just fitting parameters” (L227). If so, the Birch-Murnaghan EoS applied in this study also loses its validity because the EoS formula is based on the potential energy change by the finite strain and its parameters are resulted from the thermodynamically derivatives of the energy. As the authors suggested, it can be hard by the only present data to investigate the physically-meaningful density for the high-pressure liquid Fe-S. However, it is possible to evaluate the obtained density or its temperature when using previously reported the density data of liquid Fe-S by the authors (Nishida et al. 2011AmeMine; 2013PCM; Terasaki et al. 2019JGR; Umemoto et al. 2014JGR) and/or by the others (Morard et al. 2013EPSL; Huang et al. 2018JGR).

Indeed, it is difficult to estimate the ρ_0 value for liquid $\text{Fe}_{80}\text{S}_{20}$ based on earlier high-pressure density data. The P - ρ relations for liquid $\text{Fe}_{80}\text{S}_{20}$ previously obtained by theory (Umemoto et al., 2014), DAC (Morard et al., 2013) and shock-wave experiments (Huang et al., 2018) suggest different ρ_0 values ranging from 5.7 to 6.8 g/cm³ (see Fig. R1 below). We chose $\rho_0 = 6.2$ g/cm³ that is consistent with the shock-wave data, but we are not confident of this value.

Fig. R1 | Previous pressure–density data and inferred reference density at 1 bar for liquid $\text{Fe}_{80}\text{S}_{20}$. Green solid/open squares, density from theory²¹ without/with temperature correction; blue open/solid circles, density from shock experiment³⁹ without/with temperature correction; red squares, density from DAC experiments⁴² obtained on an adiabatic path considered here. Fitting compression curves gives $\rho_0 = 6.8, 6.2$ and 5.7 g/cm^3 for theory²¹, shock³⁹ and DAC experiments⁴², respectively.

In response to this comment, we have rewritten the main text as follows (Line 244–249);

“Here we chose the ρ_0 values for liquids $\text{Fe}_{80}\text{S}_{20}$ (ref. 39) and $\text{Fe}_{57}\text{S}_{43}$, which explain the P – ρ data obtained by shock experiments³⁹. Such ρ_0 for liquid $\text{Fe}_{80}\text{S}_{20}$ is, however, not consistent with the other P – ρ relations by theory²¹ and experiments⁴². Therefore, the ρ_0 , K_{S0} and $K_{S'}$ for liquids $\text{Fe}_{80}\text{S}_{20}$ and $\text{Fe}_{57}\text{S}_{43}$ listed in Supplementary Table 3 are not certain.”

Minor comments:

–“..., only data collected at ~13 GPa and higher...” (L99-100), how did the authors chose this value? Though the slope in Fig. S1 looks to change around ~10 GPa for $\text{Fe}_{80}\text{S}_{20}$ but at ~6 GPa, more quantitative evaluation for the critical pressure is required.

Indeed, fitting to data $\geq 10.0 \text{ GPa}$ (suggested here by the Reviewer) and $\geq 13 \text{ GPa}$ (the original pressure range for fitting) does not give a difference in extrapolated V_P beyond uncertainty (see Fig. R2 below).

By following the advice, we changed the fitting pressure range to $\geq 10.0 \text{ GPa}$ and $\geq 6.6 \text{ GPa}$ for liquids $\text{Fe}_{80}\text{S}_{20}$ and $\text{Fe}_{57}\text{S}_{43}$, respectively, in the revised manuscript.

Fig. R2 | Extrapolation of pressure–velocity curve obtained with different pressure ranges for fitting.

- In “Method”. It is well known that it is difficult to keep the liquid state under high-pressure and -temperature for long time. The authors may carefully check the liquid during measurements based on XRD from the samples. The textural image of recovered samples of Fe-S can be evident of the liquid state during ultrasonic measurements. However, it is hard to judge the liquid state of pure metal such as Fe from recovered samples. I recommend the authors to show the XRD patterns for the judgment during experiments. In addition, please describe the time scale of the ultrasonic measurements for the present liquid samples, in other words, how long the authors kept the liquid under high pressure and temperature for the measurements.

Following this suggestion, we have newly added Supplementary Fig. 6 (see below), showing our data collection in time sequence (time is given in each panel). As pointed out by the Reviewer, it is difficult to judge melting from textures of a recovered sample. The molten state of a sample is indicated by XRD diffuse signals that are characteristic of liquid. We also employed changes in ultrasonic echo signals and radiographic image to identify melting.

Supplementary Figure 6 | Data acquisition flow in run #P407. See Methods for details. Time is given in each panel. The image next to the XRD pattern shows the difference in radiographic image between the first and the second collections, indicating the distributions of molten and solid parts in partially molten sample.

In response to this comment, we have added this Supplementary Fig. 6 and the following statement in Methods (Line 182–188);

“Data acquisition flow is given in Supplementary Fig. 6. Upon each temperature increase by 100–200 K, we first obtained radiographic image and ultrasonic echo signals simultaneously, then XRD data, and collected the radiographic image and ultrasonic signals again. The molten state of a sample was indicated by XRD diffuse signals that are characteristic of liquid. We also employed changes in the radiographic image and ultrasonic signals to identify melting. We kept the molten state of a sample for 1–4 hr in total, with data collection time of 7–15 min at each temperature.”

-Please give the uncertainties for the fitting parameters in “Supplementary Table 3”.

On the basis of this comment, we added errors in K_{S0} and K'_S for Fe. However, K_{S0} and K'_S for $\text{Fe}_{80}\text{S}_{20}$ and $\text{Fe}_{57}\text{S}_{43}$ are derived by assuming ρ_0 (ρ_0 is largely unknown as we argued above). Therefore, we are not confident of these values and not willing to show their uncertainties.

- “assuming g is constant” (L243). It is unclear what values for which alloys the authors assumed here. The Grüneisen parameter has the volume and composition

dependency. Also, several values for the Grüneisen parameters among Fe-S alloys are listed in the referred papers. Please clarify which values used here.

This part related to the Grüneisen parameter and liquid isentrope was removed from the manuscript. Instead, we added brief discussion on the temperature effect on V_P , which is more relevant to argue that temperature correction on the present velocity data is not necessary.

-In Fig. S3, the symbols of "This study" and "I14" are hardly distinguished by my eyes. Please make clear diffidence among the symbols.

Changed as advised.

-In Fig. S6, i of Ni in the legend should not be subscript.

Corrected (now Supplementary Fig. 12).

-Table S1. I am worry that the authors overestimate the oxygen contamination in the recovered sample. Usually, small amount of O can be detected even pure metal due to the surface oxidation of metallic samples during the sample preparation. Did you consider such an artificial oxygen in the EPMA analysis?

We added the following statement in our revision (Line 212–214);

“Contamination by a minor amount of oxygen could have occurred during sample preparation for EPMA analysis, and therefore the oxygen content in a liquid sample may have been smaller.”

Reply to reviewer #3: (Reviewer's comment in blue italic, and our response in black)

This manuscript reports new experimental data on the sound velocity of liquid Fe and two liquid Fe-S alloys (Fe80S20 and Fe57S43) to 20 GPa. These data add to existing data at relatively low pressures, but demonstrate a different pressure dependence. The manuscript claims that if these data can be extrapolated to to Martian core pressures (20-40 GPa), then there would be little effect of S on the velocity of liquid Fe. It this is the case, then future seismic data from planetary missions may not be used to constrain the S content in the core, but may suggest the presence of light elements other than S, depending on the actual seismic velocity observations.

These are challenging experiments and the data are important to constraining the composition of the liquid outer core of Mars. However, there are some issues with the experiments as well as the data analysis as detailed below that could result in large uncertainties with the extrapolation, hence weaken the major point of the paper. These need to be clarified and additional measurements may be required to validate the results of the paper.

1. I am concerned with the consistency of the data presented in this manuscript with some of the co-author's previous data published in Terasaki et al, 2019, Journal of Geophysical Research, doi: 10.1029/2019JE005936. This manuscript does not

mention Terasaki et al. (2019) at all, but Terasaki et al. (2019) performed similar experiments on Fe-10Ni-17S and Fe-10Ni-30S up to 14 GPa, and showed a different pressure dependence of V_p for the S-rich composition than observed in this study. Although there is some difference in composition that samples in Terasaki et al. contained Ni, but as discussed Terasaki et al, Ni has little effect on the velocity. The Terasaki et al. 2019 results need to be compared in this paper, and the discrepancy needs to be discussed and clarified in the manuscript.

We compare the present velocity data with those of liquid Fe-Ni-S recently obtained by Terasaki et al. (2019) using a similar method (see Supplementary Fig. 11 below). Indeed, both data set are pretty consistent with each other when we consider larger errors in velocity in Terasaki et al. (2019), in the sense that the higher S content reduces the velocity below ~10 GPa.

Supplementary Figure 11 | Effect of Ni on V_p in liquid Fe-S at high pressure. Data for nickel-bearing liquids are from Terasaki *et al.*²⁷

In response to this comment, we added such statement in the Supplementary Information and this Supplementary Fig. 11. Additionally, we now mention the effect of silicon on the velocity of liquid Fe alloys in the Discussion section (Line 121–125), which was examined by Terasaki et al. (2019);

“For example, it has been reported^{26,27} that the velocity of liquid Fe is enhanced by the incorporation of carbon and silicon at 20–40 GPa. Carbon and silicon will therefore be alternative candidates for the light element in the Mars’ core if higher velocity is obtained. The effects of the other possible light elements such as hydrogen and oxygen remain to be explored at pressures relevant to the Martian core.”

2. The sound velocity data for the Fe-S liquids were analyzed using Eqs (2-4) without the knowledge of density. The manuscript argued that although this would constrain the equation of state, but would affect the extrapolation for V_p . This needs to be demonstrated in more detail, for example different fittings with different assumed

densities need to be performed to show the effect of the choice of density. It is potentially dangerous to leave one parameter unconstrained during the extrapolation (interpolation might be OK).

Indeed, the choice of ρ_0 does not affect the extrapolation of V_P to higher pressures. In response to this comment, we have prepared Supplementary Fig. 9 (see below). This is now mentioned in Methods (Line 233–236) as follows;

“The P – V_P fitting curve is practically independent on the choice of ρ_0 (Supplementary Fig. 9), while there is a trade-off between ρ_0 and the combination of K_{S0} and K_S' . In other words, when ρ_0 is unknown, it is difficult to obtain ρ and K_S precisely but is possible to extrapolate V_P to higher pressure.”

Supplementary Figure 9 | Effect of the choice of ρ_0 on the extrapolations of V_P to higher pressure. a, $\text{Fe}_{80}\text{S}_{20}$. b, $\text{Fe}_{57}\text{S}_{43}$.

Also, the calculation of adiabatic temperature (eq. (5)) requires the knowledge of density. The uncertainties in T_s due to the choice of density should also be discussed.

This part related to liquid isentrope (adiabatic temperature profile) was removed from the manuscript. Instead, we added brief discussion on the temperature effect on V_P , which is more relevant to argue that temperature correction on the present velocity data is not necessary.

3. The sound velocity of liquid Fe was determined using the type 2 assembly, in which the sample was fully surrounded by single crystal sapphire. Because of the hardness of single crystal sapphire, it is questionable that the sample inside would achieve the same pressure as indicated by the pressure markers that were partially surrounded by soft materials. This would result in some overestimation of the experimental pressure in these experiments, and hence could at least partially explain the discrepancy in the sound velocity of liquid Fe between this study and those from references 6, 7, and 27, 28. This need to be examined experimentally and discussed in the manuscript.

The previous work by Nishida et al. (2008) demonstrated that pressure of a sample fully surrounded by alumina became identical to that of a pressure marker above 1073 K, suggesting that this is not an issue in the present experiments. Indeed, the present data for liquid $\text{Fe}_{80}\text{S}_{20}$ using a sapphire sleeve (blue symbols) are pretty consistent with those using a BN container (red symbols) (see Fig. R3 below).

Fig. R3 | Comparison between data obtained for samples fully surrounded by single-crystal sapphire and those in a BN container.

Based on this comment, we added the following statement in Methods (Line 142–145);

“It has been demonstrated³⁴ that difference in pressure between a sample fully surrounded by alumina and a pressure marker disappeared above 1073 K. Indeed, the present data for liquid $Fe_{80}S_{20}$ using the type-1 and type-2 cell assemblies are consistent with each other.”

4. In Figure 4, it can be seen that there is clearly some effect of S on V_p at 20 GPa and 30 GPa (as large as a few percent). It is not clear to me why this much effect cannot be distinguished from future seismic observations as claimed in the paper. Some more quantitative discussion may help explain this.

As the Reviewer pointed out, sulfur concentration in liquid Fe changes its velocity to a minor extent. Nevertheless, the difference between the upper bound for the velocity of pure liquid Fe and the lower bound for that of liquid $Fe_{80}S_{20}$ is only 1%. The difference between liquids $Fe_{80}S_{20}$ and $Fe_{57}S_{43}$ is also only 6%. We suppose it is smaller than uncertainty in velocity determinations for the Martian core in the near future.

In response to this comment, we revised the relevant statement in Discussion (Line 117–118) as follows;

“However, considering uncertainty in velocity determinations for the Martian core in the near future, the velocity will not tell us its sulfur content.”

We thank valuable comments from three Reviewers, which certainly helped improve the manuscript.

Reviewers' Comments:

Reviewer #1:

Remarks to the Author:

In my view, the authors have adequately answered all issues I raised in my previous review. The manuscript may be published as is now.

Reviewer #2:

Remarks to the Author:

I find the revised version largely improved and the responses on my comments almost satisfying. I can recommend the paper to be published in Nature communications, but after some minor revisions.

The authors decided not to give any uncertainties of the fitting EoS parameters for Fe-S liquids. However, it should not be acceptable. When fitting a model function to experimental data, one should give a static error on the fitted parameter. The static error represents the deviation or statistical dispersion of the present data set from the fitting. So, it does not depend on the accuracy of the fitted value. I can expect the errors in the present fitting can be reasonable, when taking a look at fitting results in Figures. However, the authors should give the the values for easily evaluating the quality of the data-sets and fittings.

The description (lines 244-246) on the choice of ρ_0 for liquid Fe₈₀S₂₀ is hardly followed. The authors describe that the ρ_0 value of 6.2 g/cc is taken from Huang et al. (2018), however, the ρ_0 is 6.5 g/cc in Table S1 of the reference paper. From Fig. R1, I guess the authors adjusted the ρ_0 values so as to explain the shock compression density data with the present EoS (for liquid Fe₅₇S₄₃, may be based on Shaner et al.' data). Please describe clearly how to choose the ρ_0 values.

Reviewer #3:

Remarks to the Author:

The revised manuscript has fully addressed the previous comments. I have no further comments and would like to recommend this manuscript for publication in Nature Communications.

Reply to Reviewer #2 (Reviewer's comment in blue italic, and our response in black)

I find the revised version largely improved and the responses on my comments almost satisfying. I can recommend the paper to be published in Nature communications, but after some minor revisions.

The authors decided not to give any uncertainties of the fitting EoS parameters for Fe-S liquids. However, it should not be acceptable. When fitting a model function to experimental data, one should give a static error on the fitted parameter. The static error represents the deviation or statistical dispersion of the present data set from the fitting. So, it does not depend on the accuracy of the fitted value. I can expect the errors in the present fitting can be reasonable, when taking a look at fitting results in Figures. However, the authors should give the the values for easily evaluating the quality of the data-sets and fittings.

Following this suggestion, we have added uncertainties in fitting parameters for Fe-S liquids in Supplementary Table 3.

The description (lines 244-246) on the choice of ρ_0 for liquid Fe₈₀S₂₀ is hardly followed. The authors describe that the ρ_0 value of 6.2 g/cc is taken from Huang et al. (2018), however, the ρ_0 is 6.5 g/cc in Table S1 of the reference paper. From Fig. R1, I guess the authors adjusted the ρ_0 values so as to explain the shock compression density data with the present EoS (for liquid Fe₅₇S₄₃, may be based on Shaner et al.' data). Please describe clearly how to choose the ρ_0 values.

The Reviewer is right. We believe we have corrected this sentence as follows (Line 244–246);

“Here we chose the ρ_0 values for liquids Fe₈₀S₂₀ and Fe₅₇S₄₃, which explain the P – ρ data obtained by shock experiments (Huang et al., 2018).”

Reply to Editor

The second comment comes from myself: two weeks ago, the collection of first results from the InSight mission went online (<https://www.nature.com/collections/iiiiifgehfc>), maybe you could add a paragraph and discuss your results in the light of these first results?

Based on this advice, we have replaced ref. 3 with Giardini et al. (2020 Nature Geo.) and modified the main text as follows (Line 46–47). This first report focused on the Martian crust and mantle and therefore cannot be compared with our results on the core.

“The InSight mission is now in progress and has already revealed that Mars is seismically active (Giardini et al., 2020).”

bb

We thank for these comments and suggestions for improving the manuscript.